# In Vitro and In Vivo Characterisation of a Mucoadhesive Buccal Film Loaded with Doxycycline Hyclate for Topical Application in Periodontitis

**DOI:** 10.3390/pharmaceutics15020580

**Published:** 2023-02-08

**Authors:** Elena Dinte, Dana Maria Muntean, Vlad Andrei, Bianca Adina Boșca, Cristian Mircea Dudescu, Lucian Barbu-Tudoran, Gheorghe Borodi, Sanda Andrei, Adrian Florin Gal, Vasile Rus, Luciana-Mădălina Gherman, Oana Cadar, Reka Barabas, Mihaela Niculae, Aranka Ilea

**Affiliations:** 1Department of Pharmaceutical Technology and Biopharmaceutics, Faculty of Pharmacy, “Iuliu Hațieganu” University of Medicine and Pharmacy, 400012 Cluj-Napoca, Romania; 2Department of Oral Rehabilitation, Faculty of Dentistry, ”Iuliu Hațieganu” University of Medicine and Pharmacy, 400012 Cluj-Napoca, Romania; 3Department of Morphological Sciences, Faculty of Medicine, “Iuliu Hațieganu” University of Medicine and Pharmacy, 400012 Cluj-Napoca, Romania; 4Department of Mechanical Engineering, Faculty of Automotive, Mechatronics and Mechanical Engineering, Technical University of Cluj-Napoca, 400641 Cluj-Napoca, Romania; 5Department of Molecular Biology and Biotechnology, Faculty of Biology and Geology, Babes-Bolyai University, 400084 Cluj-Napoca, Romania; 6National Institute for Research and Development of Isotopic and Molecular Technologies, 400293 Cluj-Napoca, Romania; 7Department of Biochemistry, University of Agricultural Sciences and Veterinary Medicine, 400372 Cluj-Napoca, Romania; 8Department of Cell Biology, Histology and Embryology, Faculty of Veterinary Medicine, University of Agricultural Sciences and Veterinary Medicine, 400372 Cluj-Napoca, Romania; 9Experimental Centre of University of Medicine and Pharmacy “Iuliu Hațieganu”, 400349 Cluj-Napoca, Romania; 10INCDO-INOE 2000, Research Institute for Analytical Instrumentation, 400293 Cluj-Napoca, Romania; 11Department of Chemistry and Chemical Engineering of Hungarian Line of Study, Faculty of Chemistry and Chemical Engineering, 400028 Cluj-Napoca, Romania; 12Department of Infectious Diseases, Faculty of Veterinary Medicine, University of Agricultural Sciences and Veterinary Medicine, 400372 Cluj-Napoca, Romania

**Keywords:** antibiotic, bioadhesive film, drug delivery, MMP, oral cavity

## Abstract

Mucoadhesive films loaded with doxycycline hyclate (Doxy Hyc), consisting of mixtures of hydroxypropylmethyl cellulose (HPMC) E3, K4 and polyacrylic acid (Carbopol 940), were prepared by casting method, aiming to design a formulation intended for application in the oral cavity. The obtained film formulations exhibited a Doxy Hyc content between 7.52 ± 0.42 and 7.83 ± 0.41%, which had adequate mechanical properties for application in the oral cavity and pH values in the tolerance range. The x-ray diffraction studies highlighted the amorphisation of Doxy Hyc in the preparation process and the antibiotic particles present on the surface of the films, identified in the TEM images, which ensured a burst release effect in the first 15 min of the in vitro dissolution studies, after which Doxy Hyc was released by diffusion, the data presenting a good correlation with the Peppas model, n < 0.5. The formulation F1, consisting of HPMC K4 combined with C940 in a ratio of 5:3, the most performing in vitro, was tested in vivo in experimentally-induced periodontitis and demonstrated its effectiveness in improving the clinical parameters and reducing the salivary levels of matrix metalloproteinase-8 (MMP-8). The prepared Doxy Hyc loaded mucoadhesive buccal film could be used as an adjuvant for the local treatment of periodontitis, ensuring prolonged release of the antibiotic after topical application.

## 1. Introduction

Periodontal disease represents a global public health problem, with an increased prevalence of over 40% in the United States in adults aged 30 and older, who are affected by different forms of this disease [1]. In addition, the destructive effects on the dento-maxillary apparatus alter quality of life and increase the risk of conditions that have in common systemic meta-inflammation, such as cardiovascular diseases, chronic kidney diseases, metabolic syndrome, etc. [2]. Periodontal disease is a chronic polymicrobial inflammation; the most prominent bacteria that activate the destructive processes include: *Porphyromonas gingivalis*, some types of *Prevotella intermedia*, *Tannerella forsythia*, *Aggregatibacter actinomycetemcomitans*, *Fusobacterium nucleatum*, etc. [3,4]. The treatment of periodontal disease consists of controlling inflammation by removing the bacterial biofilm and stimulating the regeneration of periodontal tissues [5] and the association of antimicrobial agents with non-surgical treatment optimises the therapeutic result [6]. The topical application of antibiotics for the treatment of localised infections in the oral cavity represents an alternative solution to systemic administration [7,8]. The advantage resides in selective localised release, thus reducing systemic adverse reactions and ensuring a constant antimicrobial effect through prolonged release of the antibiotic [9,10]. The drug delivery systems designed for the oral cavity should overcome the challenges associated with the particular physiological aspects (salivary flow, the presence of mucus and mechanical forces), and also must be compatible with the structural and functional characteristics of the oral mucosa [11,12,13,14,15,16,17]. Modern trends focus on topical application of polymer-based systems, with increased interest in the development of mucoadhesive oral films [18]. Films are considered innovative approaches because they are based on patient-centered dosage forms with optimal compatibility. Mucoadhesive films for local therapy present multiple advantages: they are flexible, elastic systems, adaptable to the application site and well tolerated because they simulate the oral tissue. The bioadhesive polymers used in the film preparation can be natural or synthetic [19,20]. More recent approaches use mixtures of natural and synthetic polymers, in order to obtain films with moderate swelling properties, ensuring prolonged release for at least 6 h, increased adhesiveness and good tolerance [21,22]. Cellulose derivatives are considered suitable candidates for the formulation of oral systems due to their optimal bioadhesive properties. Hydroxypropylmethyl cellulose (HMPC) is a semi-synthetic, inert, viscoelastic polymer, with multiple roles in drug delivery systems, and used in many marketed medicinal products [23]. It is presented in the form of several grades that give the mucoadhesive systems particular characteristics, has a high glass transition temperature and good film-forming properties. Polyacrylic acid is a synthetic polymer with very good swelling capacity, which forms stable, resistant hydrogels with good mucoadhesive properties [24,25]. Doxycycline is an antibiotic belonging to the class of tetracyclines, active on numerous gram-positive and gram-negative, aerobic and anaerobic bacteria, as well as against *Mycoplasma*, *Chlamydia*, *Rickettsia* species, including some protozoa and mycobacteria [26,27,28]. In addition, Doxycycline is an inhibitor of Matrix metalloproteinases (MMPs), a significant advantage for the effective treatment of various oral diseases, including periodontitis [29,30,31].

An important challenge in the preparation of delivery systems based on doxycycline is the reduced stability of the antibiotic [32]. Doxycycline is transformed by oxidation into metacycline, and is also altered by epimerisation, to form derivatives such as 4-epidoxycycline and 6-epidoxycycline, compounds lacking the desired pharmacological activity. Oxidative degradation is enhanced by exposure to water molecules, light or elevated temperature. Studies have shown that using antioxidants to prevent oxidative degradation was ineffective; therefore, different approaches are needed to increase stability of doxycycline [7,33,34,35].

The main objective of the study was to develop a buccal mucoadhesive film intended for the local treatment of periodontitis by ensuring an increased mucoadhesion time at the application site and by the prolonged release of doxycycline in the oral cavity.

## 2. Materials and Methods

### 2.1. Materials

HPMC E3 (Methocel E3, Premium LV) and HPMC K4 (Methocel K4M Premium) were kindly obtained from Colorcon Limited (Kent, England), Carbopol 940 (C 940, B.F. Goodrich), Doxy Hyc (Sigma-Aldrich), Ethanol 95%, glycerol and propylene glycol which were purchased from Sigma-Aldrich (Darmstadt, Germany). All other reagents were of analytical grade.

### 2.2. Film Preparation

Mucoadhesive buccal films were prepared by solvent casting method. Accurately weighed quantities of HPMC E3, HPMC K4 and C 940 were dispersed in 35 mL ethanolic solution of Doxy Hyc, which was previously prepared [36]. The mixtures were stirred on a magnetic stirrer at 50 rpm for 45 min. The corresponding volumes of propylene glycol and glycerol were added (as plasticisers) and the stirring continued for 15 min.

The mixtures were kept at a temperature of 20 ± 2 °C for 24 h in conditions that avoided the evaporation of the solvent, then they were stirred for 5 min, at 50 rpm and sonicated for 10 min. A quantity of 25 g of each colloidal dispersion was poured on the flat surface, in Petri dishes with diameter of 9 cm, in two successive steps at 6 h intervals and dried at 20 ± 2 °C for 24 h, obtaining thin films after solvent evaporation. The films were detached from the flat surface, filled in sterile container, wrapped in aluminium foil and stored at 2–8 ± 0.5 °C until further determinations. The preparation was aseptically performed (laminar air flow hood), the samples being protected from light.

Initially, several blank mucoadhesive film formulations were prepared and subjected to preliminary studies, such as mechanical properties, swelling index, ex vivo mucoadhesion time. Three formulations with promising characteristics were selected, consisting of 0.40 g mixture of HPMC K4 and C 940, combined in a ratio of 5:3, 7:1 and 1:1, in which Doxy Hyc was incorporated and then they were subjected to in vitro and ex vivo studies. The other components (HPMC E3, Doxy Hyc and plasticising agents) were used in similar concentrations in the three studied formulations. The obtained process efficiency was 98.46% (SD ± 2.86). The best in vitro performing formulation was studied in vivo. The composition of the Doxy Hyc loaded prepared mucoadhesive film formulations is presented in Table 1.

### 2.3. In Vitro Characterisation

#### 2.3.1. Visual Inspection, Content Uniformity and Recovery of Doxy Hyc

The samples were analysed with a magnifying glass, in order to identify the presence of some imperfections, both after preparation and during storage.

Doxy Hyc content uniformity was performed by dissolving 1 × 1 cm pieces of film cut from five different areas of the studied film samples, in 5 mL of methanol; the samples were ultrasonicated for 30 min, centrifuged at 10,000 rpm for 5 min and then filtered through 0.45 µm filter (Millipore^®^, Burlington, MA, USA) to remove any undissolved polymers. Doxy Hyc concentrations were calculated using a calibration curve that proved to be linear between 5 and 100 µg/mL, by a validated HPLC-UV method (Agilent 1100 Series, Agilent Technology, Santa Clara, CA, USA), at 365 nm. The best separation was achieved on a Luna CN (250 × 4.6 mm, 5μm) column. The mobile phase consisted of acetonitrile and 0.1% phosphoric acid solution (45:55, *v*/*v*; t_R_ = 2 min), at a flow rate of 1 mL/min. In order to evaluate the stability of the antibiotic in the prepared films, the Doxy-Hyc content was determined during the storage and the recovery was also calculated at 6 and 12 months after preparation.

#### 2.3.2. Scanning Electron Microscopic Studies

SEM analysis of the Doxy Hyc loaded films was performed by scanning electron microscopy (Hitachi SU8320 CFEHR scanning electron microscope, Tokyo, Japan). Samples were sputter-coated with 7 nm of gold in an automatic sputter-coater from Agar (Stansted, UK).

#### 2.3.3. X-ray Scattering Measurement

X-ray patterns diffraction were collected at room temperature with a Rigaku SmartLab multipurpose diffractometer (Tokyo, Japan), using Cu Kα1 radiation (λ = 1.54056 Å), equipped with a 9 kW rotating anode. For the acquisition of the experimental data SmartLab Guidance software was used. The samples were ground using an agate pestle and mortar to a fine homogeneous powder and mounted in a sample holder. The measurements were performed in the 3–65° range in steps of 0.01°.

#### 2.3.4. pH Study

Pieces of 2 × 2 cm from the studied film samples were kept in contact with 10 mL phosphate buffer solution pH 6.8, for three hours. They were sonicated for 10 min every hour, the solutions were filtered and the pH was determined with a digital pH meter (Consort™). The results represent the average of three determinations (*n* = 3).

#### 2.3.5. Thickness, Folding Endurance, Tensile Strength

Folding endurance was determined manually by repeatedly folding the mucoadhesive buccal film samples, 180° in the same place, up to a maximum of 350 times. The experiments were repeated three times, on randomly chosen pieces of film.

The thickness of the films was measured at six different locations using a digital micrometer and the average thickness was calculated. The mechanical properties of the film samples were determined by standard tensile tests. All measurements were carried out at room temperature (23 ± 0.5 °C) for specimens with constant cross-section cutout from the prepared films. The samples were loaded to failure with a crosshead speed of 2 mm/min using a tensile test machine (Instron 3366, Norwood, MA, USA). Minimum five specimens were tested for each buccal film. The tensile strength (MPa) and elongation at break (%) were calculated, the results were reported as mean ± SD of the measurements.

#### 2.3.6. Swelling Index

The swelling index was studied to estimate the swelling behaviour of the films in contact with saliva, after application in the oral cavity. Swelling index was calculated according to some methods described in the literature [36], by weighing 1 × 1 cm pieces of mucoadhesive buccal film, before (W_1_) and after swelling (W_2_), using the formula:(1)$$\%\;\mathrm{Swelling}\;\mathrm{index}=\frac{W_{2}-W_{1}}{W_{1}}\times 100$$

The films were placed in Petri dishes, on an agar layer and left to swell in phosphate buffer solution, pH 6.8, in the climatic chamber (Venticell™) at 37 ± 1 °C. After 180 min, the swollen films were extracted from the swelling medium and reweighed, after removing the excess of water from their surface using filter paper. The results were tested in three replicates with the obtained mean result.

#### 2.3.7. In Vitro Release of Doxy Hyc from Mucoadhesive Buccal Films

In vitro release studies of Doxy Hyc were performed on mucoadhesive buccal film formulations using a USP method (Apparatus I, phosphate buffer solution, pH 6.8, 50 rpm, 37 ± 0.5 °C) (dissolution apparatus pharma test PT-DT7).

Pieces of appropriate sizes from the studied film formulations were cut and weighed, then placed in the basket and were immersed in 500 mL of dissolution media. Five milliliter samples were periodically collected and replaced with fresh dissolution medium, maintaining the sink conditions. The levels of Doxy Hyc were analyzed using the HPLC-UV method described above. All the experiments were performed in triplicate. Results are expressed as mean ± SD.

#### 2.3.8. Kinetic Release of Doxy Hyc

To understand the mechanism of Doxy Hyc release, correlation coefficients (R^2^) and release rate constants (K) for various models (zero order, first order, Higuchi and Korsemeyer–Peppas model) [37] were determined for all studied film batches by Microsoft Excel.

### 2.4. Ex Vivo Mucoadhesion Time

The time required to detach the studied films from chicken oral mucosa obtained from a local slaughterhouse was determined according to a method described by Gupta et al. [38], using an adapted tablet disintegration device (Erveka GmbH, Langen, Germany). After sacrifice, the tissue was kept at 2–8 ± 0.5 °C in sterile isotonic saline solution and the studies were carried out in the following 72 h. The adipose tissue was removed and the mucosa was fixed with cyanoacrylates on double-adhesive tape applied to a plexiglass plate. Film pieces of 1 × 1 cm were fixed on the mucosa, pressing with the finger for 5 s, and after 2 min the plate was fixed in the device in a vertical position and was moved by raising and lowering in phosphate buffer solution pH 6.8 at 37 ± 1 °C until complete separation/disintegration of the films. The average of 5 determinations was calculated.

### 2.5. In Vivo Studies

#### 2.5.1. Experiment Design

The experimental protocol for the clinical evaluation of the novel Doxy Hyc mucoadhesive buccal film was approved by the Institutional Ethics Committee of the University of Medicine and Pharmacy “Iuliu Hațieganu” Cluj-Napoca (protocol number 47/1 March 2021) and the Romanian National Sanitary Veterinary and Food Safety Authority (authorization number 266/28 June 2021). All the national and European laws regarding the ethics and safety of experimental animals were respected.

The present experiment was conducted in the Laboratory Animal Facility—Centre for Experimental Medicine, “Iuliu Hațieganu” University of Medicine and Pharmacy Cluj-Napoca, using Wistar rats as subjects. The Doxy Hyc mucoadhesive buccal film was evaluated by treating experimentally induced periodontitis and comparing the treatment outcomes to subjects with no treatment, as well as to subjects with non-surgical, mechanical treatment.

In the present study, 35 male Wistar rats, similar in age and weight, were randomised into four study groups: absolute control group—AC (*n* = 5)—subjects with no periodontal disease and no treatment); disease control group—DC (*n* = 5)—subjects with periodontal disease, but no periodontal treatment; test group 1—T1 (*n* = 15)—subjects with periodontal disease, treated by mechanical debridement and application of the Doxy Hyc mucoadhesive buccal film; test group 2—T2 (*n* = 10)—subjects with periodontal disease, treated by mechanical debridement.

The experimental protocol encompassed three phases, occurring at different time intervals. Phase 1: the selection of subjects and application of ligatures (DC, T1 and T2). Phase 2 (after 14 days): the validation of the induced periodontitis and application of treatments (T1 and T2). Phase 3 (after 28 days): the evaluation of treatment outcome (T1 and T2) and subject sacrificing, with the purpose of obtaining tissue samples for histological examination.

#### 2.5.2. Induction of Periodontitis

Local periodontitis was induced, under anesthesia, in the lower incisors during Phase 1 of the experiment using the ligature technique [39]. A silk multifilament ligature of 5/0 width was applied around the teeth, to promote plaque retention and inflammation onset. The ligatures were kept in place for 14 days; then, the ligatures were removed, and different treatments were applied to the subjects in T1 and T2 groups.

#### 2.5.3. Periodontal Treatment

In both test groups, mechanical debridement of the periodontal pockets was performed using a 1/2 Gracey curette (Hu-Friedy, Chicago, IL, USA). Additionally, in test group 1, the Doxy Hyc mucoadhesive buccal film was applied three times, at three days intervals. The buccal film was dosed by measuring trapeze-shaped samples of similar size: 10 mm width and 5 mm height. The tooth mobility and gingival index were evaluated before each application of the Doxy Hyc mucoadhesive buccal film.

The treatment efficacy was evaluated after 28 days after the first treatment administration. The subjects were then euthanised by administration of an overdose of anesthetic and the tissue specimens were collected.

#### 2.5.4. Assessment of MMP-8 Salivary Levels

Throughout the three phases of the study, saliva samples were harvested and subsequently used for the quantification of the MMP-8 concentrations, using the ELISA technique. Saliva sampling was performed under anesthesia, using intramuscularly administered mixture of Ketamine (0.2 mL 10% solution, Vetaketam, Przedsiębiorstwo Wielobranżowe VET-AGRO, Lublin, Poland) and Xylazine (0.05 mL 2% solution, Xylazin Bio, Bioveta, Ivanovice na Hane, Czech Republic) (1:1). The oral cavity of the subjects was embrocated with a mixture of citric acid and sterile saline solution, to stimulate salivation. After a few minutes, saliva was harvested in sterile Eppendorf containers, and stored at −80 °C until processing.

#### 2.5.5. Evaluation of Periodontal Parameters

The clinical aspect of the periodontium was assessed during the three phases of the study using two periodontal parameters: tooth mobility and gingival index, as proposed by Xu Y. et al. [9]. The tooth mobility was assessed with scores between 0 (lack of tooth mobility) and 3 (severe mobility). The gingival index was used to assess the texture and the colour of the gingiva, and the bleeding upon probing of the gingival sulcus. Score 0 (absence of inflammation and bleeding) was consistent with periodontal health, whereas score 3 (severe inflammation and spontaneous bleeding) was interpreted as chronic periodontitis.

#### 2.5.6. Histological Examination

The tissue samples including the lower incisors, soft and hard periodontal tissues, were harvested using a scalpel and a diamond bur, under constant saline irrigation. The samples were kept in 10% formalin solution for 4 h, and then decalcified in trichloroacetic acid for approximately four weeks. Subsequently, the samples were dehydrated in ethylic alcohol, clarified in 1-butanol, and included in paraffine blocks. Sections of 5 microns were set onto microscopic slides and coloured using Goldner’s trichrome staining method. Images were captured using a medical Leica microscope (Leica M320, Leica Microsystems GmbH, Wetzlar, Germany).

### 2.6. Statistical Analysis

Data were represented as means ± SD. Differences were considered statistically significant for values of *p* < 0.05. The student *t* test was used for the comparison of the physico-chemical properties’ values and MMP-8 salivary concentrations scores throughout the three phases of the experiment. For data analysis, dedicated software was used, including the one previously presented in the methods section, and GraphPad Prism 5, GraphPad Software (San Diego, CA, USA) for the data obtained in the in vivo studies. The ANOVA one way analysis of variance was used for the evaluation of the gingival index and tooth mobility scores throughout the three phases of the experiment.

## 3. Results

### 3.1. In vitro Characterisation

#### 3.1.1. Physical Appearance, Uniformity of Content and Stability of Doxy Hyc in Mucoadhesive Buccal Films

The Doxy Hyc loaded mucoadhesive buccal films presented a homogeneous appearance, light yellow colour, peelable and homogeneous with no evidence of drug separation upon visual inspection (Figure 1). Formulation F2, containing the highest concentration in C 940, presented a gummy, very sticky and elastic appearance (Figure 1E). No physical changes such as colour, texture and other physical parameters were observed during storage.

The content in doxy Hyc was identified between 7.52 and 7.83 mg Doxy Hyc/100 mg film, with low standard deviation, which indicates the uniform dispersion of the antibiotic in the prepared films. The concentration of over 90% of Doxy Hyc after 12 months of preparation denotes that the preparation method and the storage conditions ensured the stability of doxycycline (Table 2).

#### 3.1.2. Surface Morphology of Studied Doxy Hyc Loaded Mucoadhesive Buccal Films

The surface morphology is a widely used technique to study the sub-microscopic details of various drug carriers, including films. SEM images of the studied Doxy Hyc loaded films assessed using scanning electron microscopy are presented in Figure 1. The film surface appeared smooth and compact, with no apparent pores, suggests that all formulation constituents were mixed and uniformly distributed in the film. However, the presence of some drug particles can be observed on the surface of the films. The SEM images show irregular particles of Doxy Hyc, without well-defined shapes, which indicates the presence of the antibiotic in an amorphous state.

#### 3.1.3. X-ray Diffraction Analysis

The X-ray diffraction patterns for the compounds included in the composition of the film and for the studied film formulation, F1 (noted as Film 1) are presented in Figure 2. It can be seen that apart from Doxy Hyc, all other components are presented in amorphous form. Thus, for HPMC E3 and HPMC K4, each of them has two diffraction peaks (amorphous halo) of comparable intensity. Both of these two samples also have some very weak diffraction peaks originating from a small amount of the crystalline phase present in these samples. C940 and glycerol have one very prominent amorphous halo and one very broad at a higher angle. PG and Film 1 have each a slightly intense peak and the second maximum that appears at higher angles and has a greater intensity. The Film 1 is completely amorphous, having a broad diffraction peak and low intensity at 7.4° and another one of higher intensity at 20.6°. Doxy Hyc in its pure state is present in the crystalline phase with the characteristic diffraction lines of this compound. The diffraction peaks characteristic of Doxy Hyc disappear in the Film 1 due to the fact that it has undergone a process of amorphisation.

#### 3.1.4. pH

The pH measurement provided information on the tolerance of the films, because acidic or basic pH can be irritating to the mucosa. The studied film samples presented values between 5.9 ± 0.20 and 6.8 ± 0.27 (Table 3), a pH range of 5.5–7.0 being considered well tolerated [37].

#### 3.1.5. Thickness, Folding Endurance, Tensile Strength

The thickness values were in the range of 0.35 ± 0.01–0.45 ± 0.02 mm with low standard deviation which indicate that the obtained films are uniform. The thickness values increased in the order F3 < F1 < F2, the film with the greatest thickness being F2, which has the highest content in C 940 and the difference is statistically significant compared to the other two films (F1/F2, F3/F2, *p* < 0.05).

The folding endurance is a measure of the integrity and flexibility of the films. All the studied films did not break up to 350 performed folds, and this fact indicates appropriate properties of the films. Tensile strength is an important indicator that characterises the film’s material and its ability to withstand loads, the requirement for oral films being increased values of this parameter. The studied films presented tensile stress values in the range of 0.122 ± 0.012–0.39 ± 0.066 MPa (Table 3), a lower value being observed at F2. The values obtained are comparable to those obtained in previous studies [37] and the differences between the three formulations were statistically significant (F1/F2, F1/F3, F2/F3, *p* < 0.001).

#### 3.1.6. Swelling Index

Swelling index showed values between 11.62 and 43.47% (Table 3) and increased in the order F 1 < F 2 < F 3, the differences between formulations being statistically significant (F1/F2, F1/F3, F2/F3, *p* < 0.001). Formulation F3, consisting of a mixture of hydrophilic polymers, C 940 and HPMC K4, in a 1:1 ratio, presented the highest swelling index; this behaviour is explained by the fact that HPMC K4 is a good hydrodispersible polymer that allowed a faster swelling of C 940. Formulation F1 combining C 940 and HPMC K4 in a ratio of 5:3 presented the lowest swelling index.

#### 3.1.7. In Vitro Drug Release

The cumulative curves representing the release of Doxy Hyc from the studied mucoadhesive buccal film formulations show a prolonged release of the antibiotic during the study, and after the first 12 h Doxy Hyc was released in a ratio of 90% (Figure 3). There are no significant differences between the studied film formulations regarding the yielded percentages.

#### 3.1.8. Mathematical Modeling of Drug Release Profiles

In order to highlight the release mechanism of Doxy Hyc from the studied film formulations, the release data analysis was carried out with various mathematical models, and the correlation coefficient (R^2^) and the release exponent values (n) were used to identify the best appropriate model. Good correlation coefficients were obtained for the Higuchi model showing that the Doxy Hyc was released by diffusion from the gel layer resulting from the swelling of the hydrophilic polymers from the film composition with the dissolution medium. The analysis of the data using the Peppas model highlighted values for n < 0.5, which confirms the fact that Doxy Hyc release occurs by diffusion (Table 4).

### 3.2. Ex Vivo Mucoadhesion Time

The average time required to detach the studied mucoadhesive films from the chicken oral mucosa presented values between 180 ± 10 and 285 ± 17 min (Table 3) and increased in the order F3 < F2 < F1. Formulation F1 presented the most prolonged mucoadhesion time, differences between the studied film formulations being statistically significant (F1/F2, F1/F3, F2/F3, *p* < 0.01). The results of the in vitro studies recommended the F1 formulation for further in vivo studies.

### 3.3. In Vivo Studies

#### 3.3.1. Clinical and Histological Aspects of Experimentally-Induced Periodontitis

The ligatures were successfully applied and maintained for 14 days; none of the ligatures were removed by the subjects. Clinically, the silk ligatures induced the accumulation of food debris and the subsequent inflammation of the periodontal tissues (Appendix A).

Histologically, the lesions found in the gingiva consisted in necrosis of the gingival epithelium and the presence of abundant inflammatory infiltrate in the lamina propria, with neutrophils, macrophages, multinucleated giant cells, lymphocytes and plasma cells (Figure 4A). The inflammatory infiltrate also extended in the periodontal ligament surrounding the congested blood vessels, with associated hemorrhage. Periodontal inflammation was accompanied by intense fibroblastic reaction (Figure 4B). The alveolar bone underwent resorption, with the subsequent detachment of the ligaments from the bone surface. In the dental pulp, the blood vessels were dilated, surrounded by hemorrhages and connective tissue edema (Figure 4C). The histological diagnosis was pyogranulomatous periodontitis (Figure 4).

#### 3.3.2. Clinical Effect of Doxy Hyc Mucoadhesive Buccal Film

After removal of the ligatures, the mechanical debridement was performed in subjects in T1 and T2 groups. Subsequently, the Doxy Hyc mucoadhesive buccal film was applied to subjects in the T1 group three times: one immediately after debridement (Appendix A), three days and six days later. The application was favoured using one or two drops of sterile saline solution. Clinically, a progressive remission of the periodontal inflammation was observed during the local treatment with the buccal film. Traces of the buccal film were found three days after each application (Appendix A).

#### 3.3.3. Evolution of the Periodontal Parameters

In Phase 2, a significant increase in tooth mobility was observed in the DC group (1.6, SD ± 0.5477), T1 group (1.857, SD ± 0.5345) and T2 group (1.6, SD ± 0.5164), compared with the AC group (0.2 ± 0.4472). In Phase 3, the tooth mobility significantly decreased in groups DC (*p* < 0.01), T1 (*p* < 0.0001) and T2 (*p* < 0.0001). After treatment, the lowest tooth mobility was observed in the T1 group (0.4, SD ± 0.5164) (Figure 5).

After the induction of periodontal inflammation, a significant increase in gingival index scores was recorded in the DC group (1.6 ± 0.5477), T1 group (1.714 ± 0.6112) and T2 group (1.6 ± 0.5164), compared with the AC group. In Phase 3, the gingival index scores significantly decreased in the DC group (*p* < 0.001), T1 group (*p* < 0.0001) and T2 group (*p* < 0.0001). After treatment, the lowest mean for gingival index was recorded in the T1 group (0.4545, SD ± 0.5222) (Figure 6).

A constant decrease in the gingival index and tooth mobility scores was recorded in the T1 group throughout Phase 2 of the study, during the three successive applications of the Doxy Hyc mucoadhesive buccal film (Figure 7). After ligature removal and first buccal film application, the gingival index decreased in the T1 group from a mean score of 1.714 (SD ± 0.6112), to a mean score of 1.143 (SD ± 0.6630) at the third buccal film application. The tooth mobility decreased from a mean score of 1.857 (SD ± 0.5345) after ligature removal and first buccal film application, to a mean score of 1.214 (SD ± 0.5789) at the third mucoadhesive film application.

#### 3.3.4. Evolution of the Salivary MMP-8 Levels

The salivary MMP-8 levels (Figure 8) revealed a significant increase in the second phase of the study, in groups DC, T1 and T2. The salivary MMP-8 concentration increased in the group DC from 4.848 ng/mL (SD ± 0.9957) to 6.985 ng/mL (SD ± 1.456), in the group T1 from 4.209 ng/mL (SD ± 0.5920) to 7.630 ng/mL (SD ± 2.214) and in the group T2 from 3.891 ng/mL (SD ± 0.4725) to 6.320 ng/mL (SD ± 0.2724). After treatment, the salivary MMP-8 levels significantly decreased in the T1 group (*p* < 0.05) and the T2 group (*p* < 0.05) compared with Phase 2. In Phase 3, the mean salivary concentration of MMP-8 was significantly lower (*p* < 0.05) in the T1 group (3.780 ng/mL, SD ± 0.04406) compared with the T2 group (4.471 ng/mL, SD ± 0.6723).

#### 3.3.5. Histological Findings

After treatment, in the T1 group, the histological examination revealed that the gingival epithelium was complete, including the keratin layer on the surface. In the superficial lamina propria, isolated lymphocytes and neutrophils associated with multiple small blood vessels were identified, but without an inflammatory reaction (Figure 9A). In the periodontal ligament, discrete edema was seen in the connective tissue bordering the blood vessels. The alveolar bone regenerated and the periodontal ligaments were inserted on the bone (Figure 9B). The histological diagnosis indicated periodontal tissues with normal aspect, without detectible inflammatory lesions (Figure 9C).

In the T2 group, the histological examination indicated that the gingival epithelium had a normal structure (Figure 10A). In the lamina propria underlying the sulcular epithelium, a discrete chronic inflammatory infiltrate with lymphocytes and few neutrophils was identified (Figure 10B). The periodontal ligaments inserted on the alveolar bone. The histological diagnosis revealed periodontal tissues with normal aspect (Figure 10C).

## 4. Discussion

The obtained film formulations presented a homogeneous appearance, a pH in the tolerance range and quantification of Doxy Hyc demonstrated its uniform distribution in the film, which was recovered in a percentage of more than 90%, 12 months after preparation. This result confirms the fact that both the method used in the preparation, by which the ethanol mixture of the components was protected from light and exposed only for 48 h at a temperature of 20 ± 2 °C, and the storage of the films at 2–8 ± 0.5 °C throughout the study, ensured the physico-chemical stability of the antibiotic. The morphological analysis through SEM revealed the presence of some particles of Doxy Hyc in an amorphous state, on the surface of the film. These results were also supported by the x-ray studies, showing that Doxy Hyc passed into an amorphous state during preparation of the films, which is consistent with previously reported studies [40]. The presence of the antibiotic in an amorphous state can increase absorption due to its higher solubility compared to the crystalline state, and the presence of Doxy Hyc particles on the surface of the film could promote a beneficial, burst release effect in the initial stage. Furthermore, the efficacy would be increased by ensuring higher antibiotic concentrations at the applied local area.

These results were also confirmed through the in vitro release studies of Doxy Hyc, which are consistent with the SEM aspects. Doxy Hyc was released in a ratio of approximately 30% during the first 15 min, after which the release was prolonged due to the diffusion of Doxy Hyc from the swelled film. In addition, the release of Doxy Hyc by diffusion was also supported by the good correlation of the release data with the Higuchi and Peppas mathematical models.

Therapeutic success in the local therapy of periodontitis is ensured by a prolonged release of the antibiotic, due to an increased mucoadhesion of the film at the application site; moreover, the film should resist the constant flow of saliva and crevicular fluid, which tends to dilute the components and remove the preparation [41]. All these characteristics depend on the mechanical properties of the system.

The results regarding the mechanical characteristics of the film formulations demonstrated very good resistance to folding and tensile strength values similar to those mentioned in the literature [12], which could enable the maintenance of their integrity under the conditions in the oral cavity and in the presence of salivary flow. However, for the correct estimation of the in vivo behaviour of the films, it was necessary to correlate the results with those obtained in the swelling and ex vivo mucoadhesion studies. Thus, the formulation F3, which presented the highest tensile strength, had completely disintegrated during the 24 h of the dissolution study, while formulations F1 and F2 had completely released Doxy Hyc, but maintained their integrity upon the end of the study. In addition, the formulation F3 presented a high swelling index of 43.47% after 180 min of swelling, after which it disintegrated slowly, and also had the shortest mucoadhesion time on the mucosa according to the studied experimental conditions. These results can be explained by the behaviour of the component polymers of the film in contact with the dissolution media.

The in vitro characteristics of the prepared mucoadhesive film formulations were influenced by balancing the concentrations of the component polymers: HPMC K4 (hydrophilic-water soluble), C 940 (hydrophilic-hydrodispersible) and the insoluble polymer (HPMC E3) which provided the film matrix strength. Thus, the higher concentration of HPMC K4 in the F3 formulation favoured the faster swelling of the C 940, and also its faster disintegration, which led to shorter retention time on the mucosa. The presence of HPMC K4 in a lower ratio in the F2 formulation, but associated with a higher concentration of C 940, led to a slower swelling; however, the film was less resistant, elastic, and the gel layer formed on the surface favoured the sliding of the film on the mucosal fragment. The combination of HPMC K4 with C 940 in a ratio of 5:3 in the F1 formulation led to appropriate mechanical properties, balanced swelling, prolonged release of Doxy Hyc and increased retention time on the mucosa so that F1 was selected for further in vivo studies, in topical application, in order to control the local inflammation caused by the bacterial biofilm.

The ligature technique allowed for a successful induction of localised periodontitis in the lower incisors of the Wistar rats. The experimentally-induced periodontal disease was confirmed clinically, histologically and through the determination of salivary MMP-8 concentrations. MMP-8 has been proposed and validated as a biomarker for the early diagnosis of periodontitis, with chairside equipment developed for regular clinical usage [42,43,44].

Our results suggest that the use of the adjunctive therapy with Doxy Hyc mucoadhesive buccal film has the potential to optimise the therapeutic outcome, compared with the gold-standard treatment—the mechanical debridement. The mean scores of both the gingival index and the tooth mobility were lower in the T1 group compared with the T2 group. Moreover, the salivary MMP-8 levels were significantly lower after the application of the Doxy Hyc mucoadhesive buccal film, which is suggestive for the remission of periodontal inflammatory processes.

The histological examination indicated that the Doxy Hyc mucoadhesive buccal film was well tolerated by the gingival tissues and improved the healing process in the periodontal tissues. The film adhered to the surface of the gingiva and promoted the re-epithelialisation in the gingival epithelium; the healing process could also be optimised by preventing the formation of the bacterial biofilm. Moreover, due to the prolonged contact between the gingiva and the mucoadhesive film, and due to the composition of the film, the released Doxy Hyc could be absorbed in the gingival lamina propria to reach subantimicrobial doses in the periodontal ligament. Furthermore, since the studied buccal film reduced the salivary levels of MMP-8, it could also limit the local collagenolytic activity in the periodontal tissues by controlling the local inflammation and by reducing the local secretion of MMP-8. These favorable conditions could synergistically promote the regeneration of degraded periodontal ligaments and resorbed alveolar bone.

The Doxy Hyc mucoadhesive buccal film presents a promising clinical perspective. Its adhesive properties ensure the intimate contact with the gingival soft tissues and promote the local release of the antibiotic. Moreover, the remanence of the material, observed after three days, in subjects that have no discipline for wound care after applying the treatment, allows for a prolonged local substance release.

The use of Doxy Hyc mucoadhesive buccal film could also be effective for the therapy of other inflammatory or infectious lesions in the oral cavity. The prolonged release of antimicrobial substances could be beneficial following soft-tissue periodontal surgery or implant therapy, by applying the Doxy Hyc mucoadhesive buccal film over the sutured flaps or donor sites of autogenous soft-tissue grafts.

## 5. Conclusions

Buccal mucoadhesive films loaded with Doxy Hyc were successfully prepared by the casting method. The films obtained presented physico-chemical and mechanical properties suitable for application in the oral cavity, with a Doxy Hyc content in the range of 7.52 ± 0.42–7.83 ± 0.41%. The chosen preparation method and the storage conditions ensured the physico-chemical stability of Doxy Hyc, the recovery of this after 12 months of study being over 90%. X-ray diffraction studies highlighted the presence of Doxy–Hyc in an amorphous state in the films and the in vitro release of the antibiotic took place in a prolonged manner by diffusion; the data presented a good correlation with the Peppas model, n < 0.5. The formulation F1, consisting of HPMC K4 combined with C940 in a ratio of 5:3, presented a reduced swelling index and a prolonged ex vivo bioadhesion time, being the most performing in vitro. In vivo studies of F1 in the experimentally induced periodontitis in rats demonstrated its effectiveness in improving the clinical parameters and reducing the salivary MMP-8 levels. Thus, the studied Doxy Hyc loaded mucoadhesive buccal film can be considered an adequate system for the prolonged release of the antibiotic after topical application in the oral cavity and can be used as an adjuvant in the local treatment of periodontitis. Further human studies are needed to validate the results obtained in this study.

## 6. Patents

Patent pending with registration number A/00445/25.07.2022 at Romanian State Office for Inventions and Trademarks: ”Mucoadhesive system composition for topical release of Doxycycline in the oral cavity”.

## Figures and Tables

**Figure 1 pharmaceutics-15-00580-f001:**
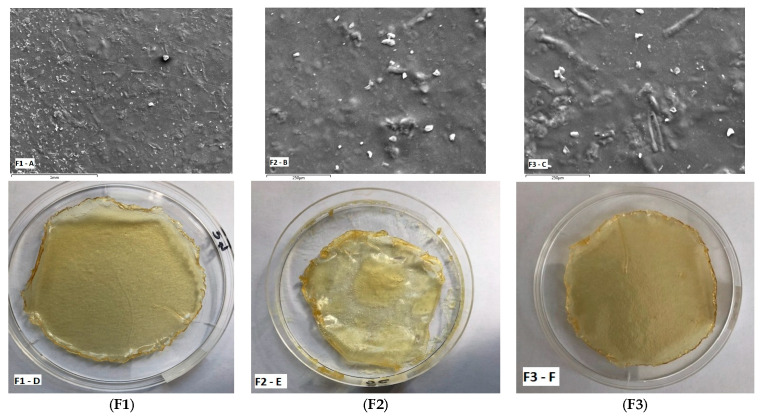
SEM images showing surface morphology (images (**A**–**C**)) and physical appearance (digital photographs) (images (**D**–**F**)) of Doxy Hyc loaded mucoadhesive buccal films ((**F1**–**F3**), according to Table 1).

**Figure 2 pharmaceutics-15-00580-f002:**
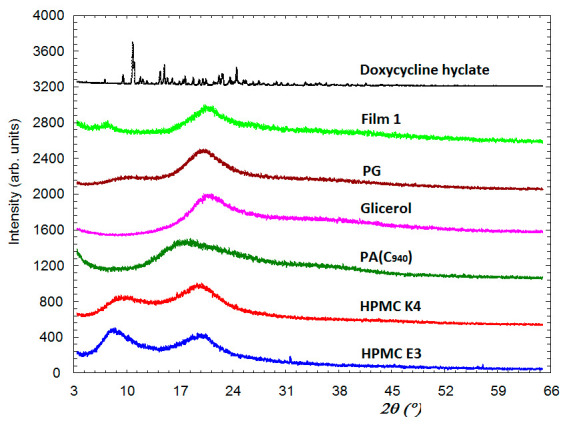
X-ray diffraction spectra of the components used in the preparation of the mucoadhesive buccal films and the F1 film formulation (according to Table 1). PG—propylene glycol; PA—polyacrylic acid (C 940).

**Figure 3 pharmaceutics-15-00580-f003:**
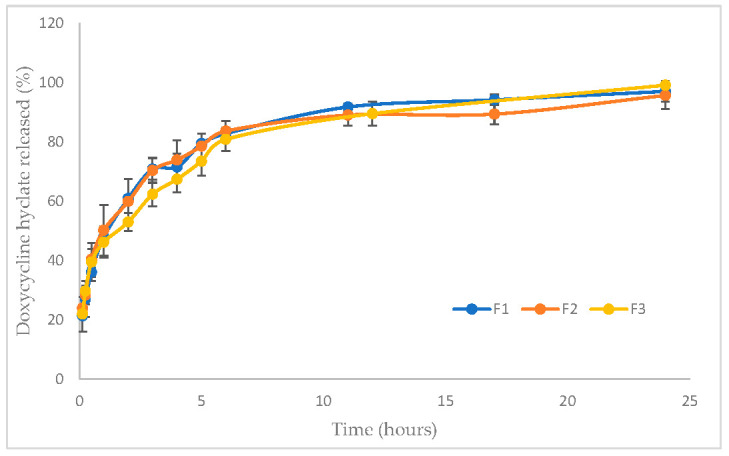
In vitro cumulative release profiles of Doxy Hyc from studied mucoadhesive buccal films (F1–F3, according to Table 1).

**Figure 4 pharmaceutics-15-00580-f004:**
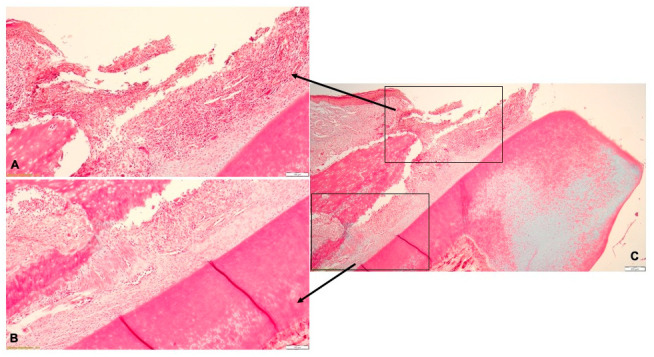
Histological aspect of the experimentally-induced periodontitis: (**A**)—necrotic lesions of the gingival epithelium and abundant inflammatory infiltrate in the lamina propria; (**B**)—resorption of the alveolar bone and detachment of the periodontal ligaments; (**C**)—pyogranulomatous periodontitis.

**Figure 5 pharmaceutics-15-00580-f005:**
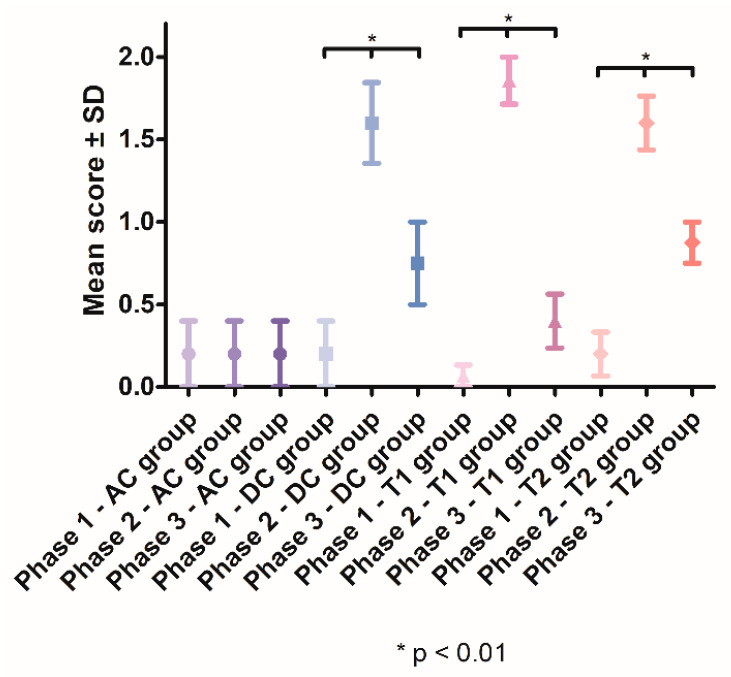
Mean tooth mobility scores recorded throughout the study.

**Figure 6 pharmaceutics-15-00580-f006:**
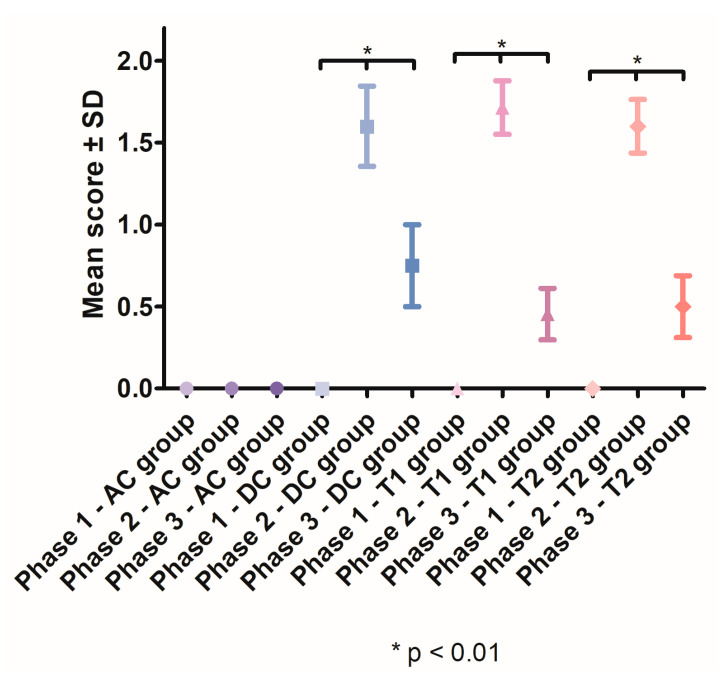
Mean gingival index scores recorded throughout the study.

**Figure 7 pharmaceutics-15-00580-f007:**
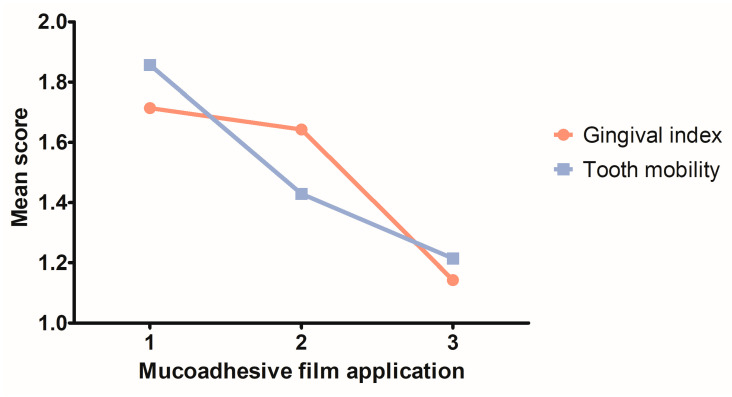
Evolution of the clinical parameters in the T1 group during the three successive topic applications of Doxy Hyc mucoadhesive buccal film.

**Figure 8 pharmaceutics-15-00580-f008:**
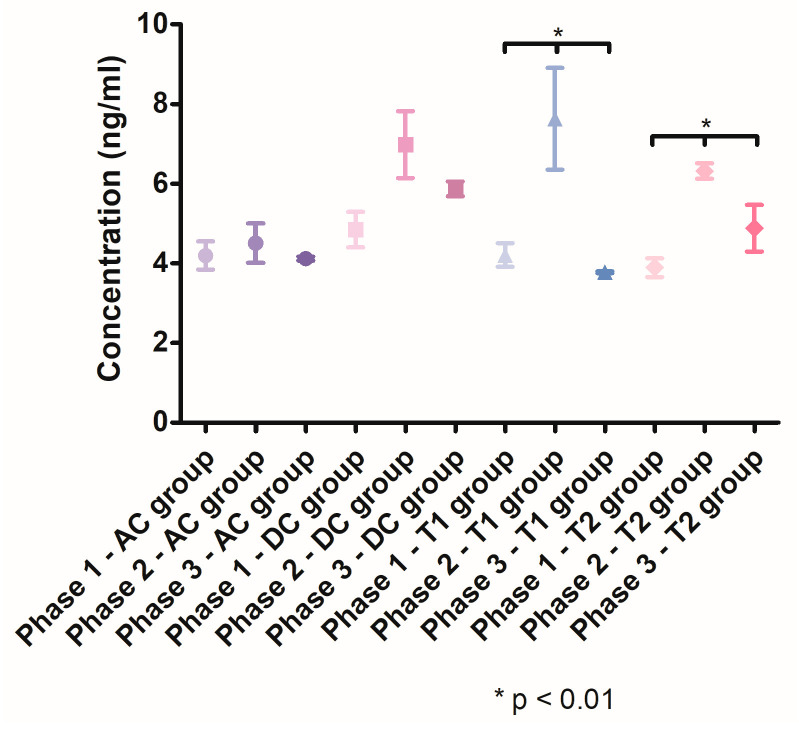
MMP-8 salivary levels throughout the three phases of the study.

**Figure 9 pharmaceutics-15-00580-f009:**
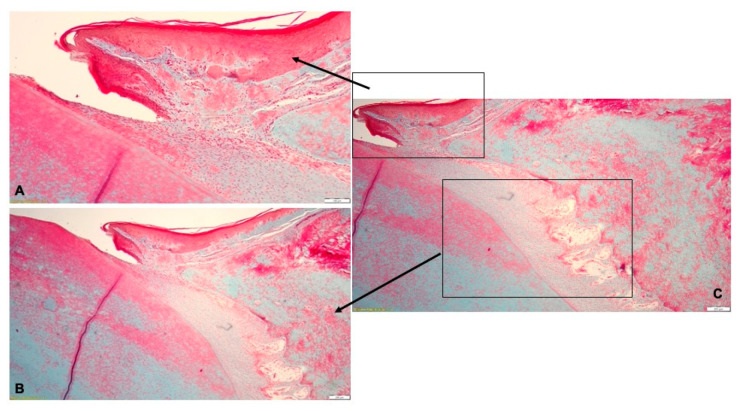
Histological aspect of T1 group after treatment with Doxy Hyc mucoadhesive buccal film: (**A**)—the gingival epithelium was continuous, and few inflammatory cells were isolated in the lamina propria; (**B**)—the periodontal ligaments inserted on the alveolar bone and discrete edema surrounding the blood vessels; (**C**)—periodontal tissues with normal aspect.

**Figure 10 pharmaceutics-15-00580-f010:**
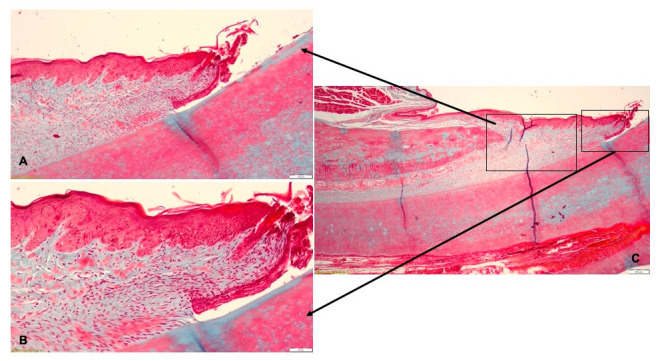
Histological aspect of T2 group after treatment by mechanical debridement: (**A**)—the gingival epithelium with normal morphology; (**B**)—discrete chronic inflammatory infiltrate in the gingival lamina propria; (**C**)—periodontal tissues with normal aspect.

**Table 1 pharmaceutics-15-00580-t001:** Composition of the prepared mucoadhesive buccal films.

Component	Formulation		
	F1	F2	F3
HPMC E3 (g)	1	1	1
HPMC K4 (g)	0.15	0.05	0.20
C 940 (g)	0.25	0.35	0.20
Propylene Glycol (mL)	1.10	1.10	1.10
Glycerol (mL)	0.20	0.20	0.20
Doxy Hyc (g)	0.25	0.25	0.25
Ethanol 95% (mL)	35	35	35

**Table 2 pharmaceutics-15-00580-t002:** The amount of Doxy Hyc loaded in mucoadhesive buccal films (F1–F3, according to Table 1) and its recovery after 6 and 12 months of storage.

Content	Formulation		
	F1	F2	F3
Doxy Hyc content in the fresh films (mg Doxy Hyc/100 mg film)	7.83 ± 0.41	7.52 ± 0.42	7.79 ± 0.39
Recovery in Doxy Hyc content in the films, 6 months after preparation (%)	94.21 ± 1.14	95.34 ± 4.22	96.25 ± 2.33
Recovery in Doxy-Hyc content in the films, 12 months after preparation (%)	93.45 ± 3.24	92.88 ± 5.23	94.65 ± 5.36

**Table 3 pharmaceutics-15-00580-t003:** Physicochemical characteristics of studied mucoadhesive buccal films (F1–F3, according to Table 1).

Parameters	Formulation		
	F1	F2	F3
pH	6.3 ± 0.33	5.9 ± 0.20	6.8 ± 0.27
Folding endurance	>350	>350	>350
Thickness (mm)	0.37 ± 0.01	0.45 ± 0.02	0.35 ± 0.01
Tensile strength (MPa)	0.232 ± 0.011	0.122 ± 0.012	0.390 ± 0.066
Swelling index (%) at 180 min	11.62 ± 3.54	17.02 ± 2.16	43.47 ± 8.27
Ex vivo bioadhesion time (min)	285 ± 10	225 ± 12	180 ± 17

**Table 4 pharmaceutics-15-00580-t004:** Kinetic models of Doxy Hyc release from studied mucoadhesive buccal films (F1–F3, according to Table 1).

Kinetic Models	Parameters	Formulation		
		F1	F2	F3
Zero—order model	R^2^	0.6193	0.5998	0.7412
	K_0_	2.76	2.53	2.94
First—order model	R^2^	0.9278	0.8799	0.9908
	K_1_	0.134	0.109	0.178
Higuchi model	R^2^	0.8434	0.8261	0.9212
	K_H_	16.83	15.46	17.05
Korsmeyer-Peppas model	R^2^	0.9589	0.9532	0.9792
	K_HP_	44.67	46.65	43.90
	n	0.303	0.275	0.286

Where, R^2^ represents the correlation coefficients. The k_0_, k_1_, k_H_, k_HP_, are rate constants following the zero-order, first-order, Higuchi and Korsmeyer–Peppas mathematical models, respectively. The n was the release exponent. The data fitting are presented in Appendix A.

## Data Availability

Not applicable.

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
