# Peer review of "In Vitro and In Vivo Characterisation of a Mucoadhesive Buccal Film Loaded with Doxycycline Hyclate for Topical Application in Periodontitis"

_pharmaceutics, 2023, doi:10.3390/pharmaceutics15020580_

Round 1
Reviewer 1 Report
This paper provides a useful addition to the scientific literature on drug delivery system. The paper is well written, clear and the discussion supports the presented data.
Author Response
Dear sir/madam,
We would like to thank you for the revision of our submitted paper to the special issue in the journal Pharmaceutics, Special Issue "Pharmaceutical Formulations with Antimicrobial Properties (Volume II)".

Reviewer 2 Report
The study entitled "In vitro and in vivo characterisation of a mucoadhesive buccal film loaded with doxycycline hyclate for topical application in periodontitis” describes the develop a buccal mucoadhesive film intended for the local treatment of periodontitis by ensuring an increased mucoadhesion time at the application site and by the prolonged release of Doxycycline in the oral cavity.
The study is very interesting and described with very good scientific rigor; only minor changes need to be addressed.
Abstract: Add the study aim.
Keywords: should be in alphabetical order, KEYWORDS should not contain the same words that are within the title of the text. Thus these should be changed appropriately.
Author Response
Dear sir/madam,
We would like to thank you for the revision of our submitted paper to the special issue in the journal Pharmaceutics, Special Issue "Pharmaceutical Formulations with Antimicrobial Properties (Volume II)".
After analysing your comments and suggestions, the following modifications have been made:
The purpose of the study was mentioned in the abstract.
Mucoadhesive films loaded with Doxycycline hyclate (Doxy Hyc), consisting of mixtures of hydroxypropylmethyl cellulose (HPMC) E3, K4 and polyacrylic acid (Carbopol 940), were prepared by casting method, aiming to design a formulation intended for application in the oral cavity. The obtained film formulations exhibited a Doxy Hyc content between 7.52 ± 0.42 – 7.83 ± 0.41%, adequate mechanical properties for application in the oral cavity and pH values in the tolerance range. The X-ray diffraction studies highlighted the amorphization of Doxy Hyc in the preparation process and the antibiotic particles present on the surface of the films, identified in the TEM images, ensured a burst release effect in the first 15 minutes of the in vitro dissolution studies, after which Doxy Hyc was released by diffusion, the data presenting a good correlation with the Peppas model, n < 0.5. The formulation F1, consisting of HPMC K4 combined with C940 in a ratio of 5 : 3, the most performing in vitro, was tested in vivo in experimentally-induced periodontitis and demonstrated its effectiveness in improving the clinical parameters and reducing the salivary levels of matrix metalloproteinase-8 (MMP-8). The prepared Doxy Hyc loaded mucoadhesive buccal film could be used as an adjuvant for the local treatment of periodontitis, providing prolonged release of the antibiotic after topical application.
Keywords: should be in alphabetical order, KEYWORDS should not contain the same words that are within the title of the text. Thus these should be changed appropriately.
The keywords were changed and were put in alphabetical order:
antibiotic , bioadhesive film, drug delivery, MMP, oral cavity

Reviewer 3 Report
The work conducted by Dinte et al. is promising and will definitely benefit the scientific community. The authors did an excellent job. However, some major concerns should be addressed before this paper can be accepted for publication; therefore, I recommend major revisions as follows;
1. In section 2.2., did the author invent this method? If not, please add an appropriate citation.
2. In line 116, please add a suitable citation.
3. The authors should conduct and present the encapsulation (entrapment) efficiency and drug loading % tests to estimate the amount of successfully loaded Doxycycline hyclate in the designed mucoadhesive films.
4. In addition, the Process efficiency (PE) should be studied (the amount of systems that can be obtained with respect to the initial amount of materials present in the formulation).
5. Previous studies reported the pH of saliva in case of periodontitis, and it was reported to be in the range 6.3 to 7.4. On the other hand, the pH of normal saliva is 7.9. I suggest conducting the in vitro release study on different pHs within this range.
6. The authors should test the biocompatibility of the designed films on normal cells using MTT assay to investigate their safety on the buccal cavity cells.
7. Please add the pharmacokinetic plots in the supplementary materials.
8. Please move Figures 4 and 6 to supplementary materials.
9. The authors should test the antibacterial effects of Doxycyclate hyclate on the bacteria causing periodontitis (Porphyromonas gingivalis, Prevotella intermedia, Tannerella forsythia, etc.) before and after loading into the mucoadhesive films since they are the leading cause of this disease.
Author Response
Dear sir/madam,
We would like to thank you for the revision of our submitted paper to the special issue in the journal Pharmaceutics, Special Issue "Pharmaceutical Formulations with Antimicrobial Properties (Volume II)".
After analysing your comments and suggestions, the following modifications have been made:
1.In section 2.2., did the author invent this method? If not, please add an appropriate citation.
The literature cites the preparation of films by casting method using different solvents (water, ethanol, propylene glycol - isopropyl alcohol mixture, etc.). The authors adapted the method according to Ammar H.O. et al., 2017 (reference number 36) using ethanol, to maintain the stability of doxycycline.
In previous studies, we used water as a dispersion medium, but the longer exposure time of the mixtures at ambient temperature, for drying, led to the degradation of doxycycline.
2.In line 116, please add a suitable citation.
Appropriate citation has been added (see line 120)
3.The authors should conduct and present the encapsulation (entrapment) efficiency and drug loading % tests to estimate the amount of successfully loaded Doxycycline hyclate in the designed mucoadhesive films.
Doxycycline hyclate was not encapsulated in any nanosystem, it was dispersed in ethanol, together with polymers and plasticizing agents, and then the ethanol was removed by evaporation.
4.In addition, the Process efficiency (PE) should be studied (the amount of systems that can be obtained with respect to the initial amount of materials present in the formulation).
The obtained process efficiency was 98.46% (SD ± 2.86). The result was mentioned in the text. (see line 137).
5.Previous studies reported the pH of saliva in case of periodontitis, and it was reported to be in the range 6.3 to 7.4. On the other hand, the pH of normal saliva is 7.9. I suggest conducting the in vitro release study on different pHs within this range.
The pH of the saliva varies during a day, depending on the nature of the food, but also depending on the physiological and pathological characteristics of the subject. The in vitro release study was carried out in phosphate buffer solution, pH 6.8, the average value accepted in the literature as being relevant for the characterization of preparations applied in the oral cavity (Gilhotra RM et al., 2015, Li A et al., 2021 (reference 18), Patlolla VGR et al., 2019 (reference 35), Khan G et al., 2016 (reference 40)).
In addition, Doxycycline hyclat is an adduct of doxycycline hydrochloride, with a better solubility in water, but also with a better physico-chemical stability. A difference of 1 unit in the pH value would not significantly influence the release rate; in the case of the studied films, the release rate is governed by the swelling capacity of the film-forming polymer mixture, but also by the viscosity of the resulted hydrogel, which influences the dissolution rate of doxycycline and its diffusion in the dissolution medium.
6.The authors should test the biocompatibility of the designed films on normal cells using MTT assay to investigate their safety on the buccal cavity cells.
This is a welcomed recommendation. We propose to carry out a biocompatibility study of the obtained film, on human periodontal cells, before the initiation of the in vivo study in humans.
7.Please add the pharmacokinetic plots in the supplementary materials.
The plots representing the fit of in vitro release data with the Higuchi and Peppas models have been added in supplementary materials. Some corrections have been made in Table 4 because the plots were newly generated, using of Microsoft Excel program.
8.Please move Figures 4 and 6 to supplementary materials.
The Figures 4 and 6 were moved to the supplementary materials.
9.The authors should test the antibacterial effects of Doxycyclate hyclate on the bacteria causing periodontitis (Porphyromonas gingivalis, Prevotella intermedia, Tannerella forsythia, etc.) before and after loading into the mucoadhesive films since they are the leading cause of this disease.
The antibacterial effect was evaluated by the favorable evolution of the clinical parameters revealed by the experimental studies in rat. The literature mentions the effectiveness of doxycycline on strains of bacteria that form the oral biofilm, including periodontopathogenic bacteria (Patlolla VGR et al., 2019 (reference 35), Bostanci N. et al., 2012, Lim SY et al., 2020, Senel S. et al., 2021).
In addition, Doxycycline is clinically approved for MMP inhibition by the US Food and Drug Administration (FDA) and its inhibitory activity on MMPs is nonspecific (Patlolla VGR et al., 2019 (reference 35)).

Reviewer 4 Report
1- In table 1, please justify on what basis did you choose the amounts of different components of each formulation.
2- Please mention the sink condition of the drug in the materials and methods of drug release.
3- Release of the free drug should be done and compared with the release of the drug from the films.
4- In Figures 11 and 12, the three images should be equal in size.
Author Response
Dear sir/madam,
We would like to thank you for the revision of our submitted paper to the special issue in the journal Pharmaceutics, Special Issue "Pharmaceutical Formulations with Antimicrobial Properties (Volume II)".
After analysing your comments and suggestions, the following modifications have been made:
1- In table 1, please justify on what basis did you choose the amounts of different components of each formulation.
In the preliminary studies, combinations in different proportions of the bioadhesive polymers were used. The study continued with three more relevant formulations, in order to select a formulation for the in vivo studies. Formulations that presented a high swelling index (those containing high concentration of Carbopol), or a reduced ex vivo retention time were eliminated.
This was already mentioned in the text:
“Initially, several blank mucoadhesive film formulations were prepared and subjected to preliminary studies, such as mechanical properties, swelling index, ex vivo mucoadhesion time. Three formulations with promising characteristics were selected, consisting of 0.40 g mixture of HPMC K4 and C 940, combined in a ratio of 5 : 3, 7 : 1 and 1 : 1, in which Doxy Hyc was incorporated and then they were subjected to in vitro - ex vivo studies”
2- Please mention the sink condition of the drug in the materials and methods of drug release.
Sink conditions were mentioned in text (see line 202).
3- Release of the free drug should be done and compared with the release of the drug from the films.
Doxycycline hyclate is homogeneously dispersed in the dry mixture of film forming polymers. Antibiotic particles are presented in amorphous form (result confirmed by XRD and TEM studies). In contact with a dissolution medium, the polymeric mixture swells, and the antibiotic will be released after dissolution.
4- In Figures 11 and 12, the three images should be equal in size.
Figures 5, 11 and 12 presenting the histological findings were modified, becoming Figures 4, 9 and 10. For each of these figures, image C is the overview, and images A and B are details, taken under a higher magnification. The size of images A and B was increased, for a better visualisation, and in image C, the rectangular inserts indicate the zones that were zoomed in the detailed images A and B.

Round 2
Reviewer 3 Report
The authors responded to most of the reviewers' comments and I recommend the acceptance of the paper in its current form.